# A Multiplexed Urinary Biomarker Panel Has Potential for Alzheimer’s Disease Diagnosis Using Targeted Proteomics and Machine Learning

**DOI:** 10.3390/ijms241813758

**Published:** 2023-09-06

**Authors:** Jenny Hällqvist, Rui C. Pinto, Wendy E. Heywood, Jonjo Cordey, Alexander J. M. Foulkes, Catherine F. Slattery, Claire A. Leckey, Eimear C. Murphy, Henrik Zetterberg, Jonathan M. Schott, Kevin Mills, Ross W. Paterson

**Affiliations:** 1Translational Mass Spectrometry Research Group, Genetics and Genomic Medicine, UCL Great Ormond Street Institute of Child Health, London WC1N 1EH, UK; j.hallqvist@ucl.ac.uk (J.H.); kevin.mills@ucl.ac.uk (K.M.); 2Faculty of Medicine, School of Public Health, Imperial College London, London SW7 2BX, UK; 3National Hospital for Neurology and Neurosurgery, Queen Square London, London WC1N 3BG, UK; 4Darent Valley Hospital, Dartford DA2 8DA, UK; 5Dementia Research Centre, UCL Queen Square Institute of Neurology, London WC1N 3BG, UK; 6Department of Psychiatry and Neurochemistry, Institute of Neuroscience and Physiology, The Sahlgrenska Academy at the University of Gothenburg, S-431 80 Mölndal, Sweden; 7UK Dementia Research Institute, UCL, London WC1E 6BT, UK

**Keywords:** Alzheimer’s, urine, machine learning, biomarkers, proteomics, mass spectrometry, diagnosis

## Abstract

As disease-modifying therapies are now available for Alzheimer’s disease (AD), accessible, accurate and affordable biomarkers to support diagnosis are urgently needed. We sought to develop a mass spectrometry-based urine test as a high-throughput screening tool for diagnosing AD. We collected urine from a discovery cohort (n = 11) of well-characterised individuals with AD (n = 6) and their asymptomatic, CSF biomarker-negative study partners (n = 5) and used untargeted proteomics for biomarker discovery. Protein biomarkers identified were taken forward to develop a high-throughput, multiplexed and targeted proteomic assay which was tested on an independent cohort (n = 21). The panel of proteins identified are known to be involved in AD pathogenesis. In comparing AD and controls, a panel of proteins including MIEN1, TNFB, VCAM1, REG1B and ABCA7 had a classification accuracy of 86%. These proteins have been previously implicated in AD pathogenesis. This suggests that urine-targeted mass spectrometry has potential utility as a diagnostic screening tool in AD.

## 1. Introduction

Alzheimer’s disease (AD) is a progressive neurodegenerative disease and the most common cause of dementia. A definite diagnosis of AD can only be made post-mortem and by neuropathological confirmation of extracellular amyloid plaques. These plaques are composed of fibrillar amyloid and intracellular tau tangles, containing hyperphosphorylated tau [1].

Since the brain is inaccessible during life, the AD field has relied on imaging and cerebrospinal fluid (CSF) biomarkers to support clinical diagnosis, classify individuals for research studies or to interrogate and track pathophysiology [2,3]. With the development of new disease-modifying therapies, it has become critically urgent to develop biomarkers that will identify individuals who are most likely to benefit from them. Thus, we need to develop tests that are simple to obtain, less invasive, high throughput and cost effective so we can test widely.

Currently, several fluid biomarkers are now validated to support clinical diagnosis in CSF [4]. These include the use of the ratio of the beta-amyloid peptides Aβ1-42/1-40, P-tau a marker of tau pathology, and neurofilament light (NfL), a biomarker of neuroaxonal damage [5,6]. There are also promising biomarkers of amyloid, tau and neurodegeneration in plasma [7] and they are in the process of being validated in real life populations, with plasma P-tau 217 looking particularly promising [8,9,10,11]. However, all these tests are expensive to perform, require invasive procedures, have to be pre-handled and centrifuged at source, and analyses performed in specialist laboratories.

CSF is obtained by lumbar puncture (LP), which is safe and generally well tolerated. However, it is not possible to perform this procedure on everyone, for instance those on blood-thinning agents, and is an invasive procedure that carries a 5–10% risk of post-LP syndrome [12]. Although blood is more accessible, it still remains invasive, and requires strict pre-handling practices with rapid centrifugation at source, to provide reliable results [13]. Therefore, there is considerable interest and a need for developing biomarkers in novel and non-invasive biological fluids such as urine that are easier and cheaper to collect, store, process and analyse. This could potentially be collected in the home environment and shipped to a lab through the post with no sample preparation required. Sampling is likely to be more acceptable to patients and could be widely used for earlier screening in mid-life populations outside of specialist centres and in any geographical location. 

Despite extensive progress in CSF and serum biomarkers, research into biomarkers for brain diseases in urine is lagging behind. This is due to the significantly lower protein concentrations making detection more difficult and that urine is more distant from the brain compared with CSF and plasma. However, with technological advances in mass spectrometry, the increasing use of triple quadrupole mass spectrometers in chemical pathology and their unique ability to perform targeted proteomic analyses of multiple biomarkers at once, it is now conceivable that urinary biomarkers of AD can be measured. Urine has been explored previously for its potential for AD diagnosis particularly for metabolite changes [14,15,16,17,18]; however, the urinary metabolome can be highly variable so we have looked at the potential of urinary proteins. Conventionally most urinary protein is thought to be kidney derived but it has been shown to contain many membrane and extracellular proteins not just from the renal system but also from other distant organs [19,20]. More recently its potential is being explored for neurological conditions such as brain cancer [21], Parkinson’s disease [22] and AD [23,24,25,26]

Furthermore, the use of machine learning and AI approaches is revolutionising diagnostics as we move to use multiple biomarkers to refine prognosis and diagnosis [27]. Therefore, in this study we tested the hypothesis that individuals with AD would have a different urinary proteomic profile to healthy controls and that these could reflect changes in relevant pathobiological pathways. The aims of this exploratory study were to:
Screen the urine proteome in participants with AD and compare with healthy controls using untargeted proteomics to identity pathways and potential AD biomarkers;Create a translational and rapid test to validate any biomarkers using a multiplexed, targeted proteomic approach in an independent cohort;Use machine learning techniques to develop a ‘panel’ of biomarkers that could be used to help diagnose AD.

## 2. Results

As expected, a significantly greater proportion of AD individuals in the discovery cohort were APO ε4-positive compared to controls in both cohorts. CSF amyloid concentration was lower in those with AD, and T-tau and P-tau were higher. MMSE scores were unsurprisingly lower in the AD group. These individuals were in the mild to moderate stage of the disease. The discovery and validation cohorts are described in full in Table 1. 

### 2.1. Biomarker Discovery and Pathway Analyses: Comparison of Urinary Proteomic Analyses from AD Patients and Healthy Controls

The discovery cohort was composed of eleven individuals: six participants fulfilled clinical criteria for typical AD, and five were healthy controls. Label free proteomics and bioinformatic analyses identified a total of 1525 proteins. Comparison between the AD and control patient group identified a total of 42 proteins demonstrating changes in expression of protein between both groups (see Appendix A for details). As described in the methods section, all proteins ranked A, B and C and demonstrating changes in expression, designated as showing a statistical significance of *p* < 0.05, were analysed using the Ingenuity Pathway analyses bioinformatics package (Qiagen, Venlo, The Netherlands).

Ingenuity canonical pathways analyses identified proteins activated by oxysterol ligands, cholesterol and bile acid metabolism (LXR/FXR/RXR activation) to be significantly changed in the urine of AD patients. Other changes observed included those involved in protein synthesis or folding and also changes in energy metabolism. Remarkably, disease function analyses identified nine proteins previously described as being altered in AD and amyloidosis diseases (APOA4, HSPA5, RNASET2, PLD3, FRMD4B, B2M, HPX, SERPINC1, OLR1). The functional pathway amyloidosis and familial amyloidosis being the first and third most significant pathways changed, with AD being the second and eighth most significantly relevant disease identified for each pathway, respectively. Figure 1 shows a summary of the results from the discovery phase.

### 2.2. Development of a Targeted and Multiplex Assay to Validate Potential Urinary Proteomic Biomarkers of AD

Unlike the pathway analyses of potential disease-modifying proteins in AD where all significant, differentially expressed proteins are taken into account, the development of a validatory and high-throughput test requires only those biomarkers with high confidence identification being taken forward. Figure 2 shows the 29 high-confidence proteins identified in this Rank A group that were taken forward for evaluation from the discovery analyses. These potential biomarkers were augmented into our in-house neuroinflammatory targeted proteomic panel [28], which was constructed from a mixture of literature reviews, known biomarkers of neurodegeneration and inflammatory factors. Therefore, a total of 88 protein biomarkers were developed into a scheduled, 17 min, multiplexed and targeted proteomic test. This panel of potential hypothesis-driven and hypothesis-generated biomarkers were then analysed in a total of 21 patients which constituted of 9 individuals with AD and 12 healthy, age-matched controls.

### 2.3. Individual Biomarkers and Univariate Analyses for the Diagnosis of AD

Out of the total of 88 biomarkers analysed in the multiplexed validatory test, we could detect reliably 61 proteins. We applied elastic net regression to pinpoint the most influential proteins in the discrimination between AD and healthy controls. We identified a total of five proteins which demonstrated potential for being put forward as possible biomarkers for AD (Figure 3a). These included the four proteins, REG1B, VCAM1, TNF-beta and MIEN1, which were observed to be present in lower concentrations in AD urine. The protein, ABAC7, was the only potential biomarker to be elevated, with a 1.5-mean fold increase in concentration in the AD group compared to the controls (Figure 3a,b). However, of all the biomarkers only MIEN1 demonstrated a statistically significant change in concentration compared to the control group (Figure 3b). The biomarkers ABAC7 and MIEN1 (FC = −1.8) which demonstrated the greatest fold-change relative to one another, was further evaluated by looking at their relationship to one another in each patient. By ratioing the biomarkers MIEN1/ABAC7, we were able to further increase the statistical significance between the AD and controls even further (Figure 3c, *p* = 0.01).

### 2.4. Multilinear Regression and Machine Learning Analyses of Multiple Biomarkers for Improving the Specificity and Sensitivity for Diagnosing AD

To improve accuracy and sensitivity of the assay, we modelled the proteins using linear regression (Figure 4). The AD and control samples were initially assessed using the significantly different protein MIEN1 alone; prediction of the samples in the model demonstrated an overall classification accuracy of 71% (56% sensitivity). To evaluate if the classification strength could be improved by using more predictors, we included the proteins which had been chosen in the elastic net feature selection performed to find the most relevant proteins to discriminate between AD and controls. This rendered us with a model including MIEN1, TNFB, VCAM1, REG1B and ABCA7. These five proteins were applied as predictors in a multiple linear regression model and when the samples were once more predicted, the overall classification accuracy was 86%, demonstrating a 15% improvement as compared to using the protein MIEN1 alone. Given the small sample size, we opted for performing k-fold cross validation on the model, where we utilised ten splits of the data. This resulted in six out of nine samples being accurately predicted as AD, and eight out of twelve samples accurately predicted as control.

## 3. Discussion

In this exploratory study, we used untargeted proteomic profiling to identify proteins that were differentially expressed in affected AD patients compared to unaffected control urine. We were able to detect differences in a number of proteins known to be implicated in AD pathology and related biological pathways. Furthermore, we developed a targeted proteomics panel and validated our findings in an independent cohort. Our findings suggest that this approach could have potential as a test in clinical practice for first tier screening or potentially to monitor new treatments. Since many of the proteins we identified are implicated in brain amyloidosis, a pathophysiological process which predates symptom onset in AD by ~25 years [29], we postulate that this biomarker panel has potential as an early pre-symptomatic screening tool for AD.

Finding biomarkers for early and accurate diagnosis of AD has become an urgent priority in the dementia field, now that three immunotherapies against fibrillar amyloid have demonstrated utility in slowing cognitive progression in mild to moderate AD [30,31]. Inclusion in these studies requires demonstration of amyloidosis and tauopathy in the brain, either by cerebrospinal fluid analysis or PET scanning, and this is likely to remain the gold standard for providing accurate individual biomarkers support for a diagnosis of AD. However, the infrastructure for carrying out LPs and PET scans in the general population is very limited [32]. Tools for screening and risk stratification in the general population will be extremely important, particularly in countries without access to advanced diagnostics for dementia [33,34]. Blood biomarkers are currently reasonably advanced in their development, with plasma P-tau 217 (particularly the P-T217/T217 ratio [35,36]) proving to be a strong predictive marker of amyloid burden and tau deposition, in real life populations [37,38] and plasma Aβ42/40 ratio allowing for reasonable separation of AD from non-AD individuals [39]. Yet, one limitation of blood biomarkers is the need for strict pre-handling protocols so that samples are centrifuged rapidly to avoid the effects of proteases degrading proteins of interest [40]. This is likely to limit its use in regional memory clinics and geographically isolated healthcare environments. It is also likely to be a barrier for home self-testing. In this study, we collected urine from a mid-stream sample. No early aliquoting or centrifugation was required at the point of collection, and very limited instructions were provided to participants, even those with moderately severe cognitive impairment. This means that collection is likely to be feasible across populations in developed and developing countries, making it an attractive screening tool and urine should be considered a more ‘liquid gold’ biofluid rather than just a waste product.

Using a univariate analysis only one protein, migration and invasion enhancer 1 (MIEN1), demonstrated a clear and statistically significant change in expression between the AD and control groups. MIEN1 is an intracellular protein located to the cytosol and centriole, that is expressed in many tissues but has a particularly high expression in the basal ganglia and cerebellum of the brain. MIEN1 is known to be a negative regulator of apoptosis and a positive regulator of cell migration and has gained significant interest in its role in breast cancer. However, MIEN1 variants have also been associated with early onset AD [41]. The exact mechanism of action is unclear, but MIEN1 is postulated to both interact with glutathione peroxidase and plays a significant role in the regulation of apoptosis through control of caspase 3 (CASP3). Variants in both these proteins are known to be associated strongly with AD, in particular CASP3 which cleaves beta amyloid 4a protein which is strongly associated with neuronal cell death observed in AD patients. Although individual statistical significance was not achieved with the other elevated proteins identified in this study, we were able to demonstrate that the diagnostic classification accuracy increases by 15% when utilising a panel of five proteins as compared to one protein alone. This highlights the strength of a carefully selected biomarker panel to improve the diagnostic and predictive ability of discriminant models. Our study is limited by the small sample size, but we anticipate that with increased sample numbers and statistical power, we could improve classification accuracy. Further, the five predictor proteins are involved in a range of biological processes that are plausibly relevant to AD pathophysiology, generally implicating lipid metabolism and inflammation pathways. Metabolomic-wide studies (MWAS) have demonstrated utility in elucidating the molecular mechanisms by which individual genes confer risk for AD. A recent study demonstrated that variations in *ABCA7* are linked to increased risk of AD through altered sphingolipid metabolism [42]. Using disease and functional analyses, we have identified changes in a number of biomarkers involved in amyloidosis and AD including PLD3. Importantly, a previous urine proteomics study found that PLD3 is associated with AD [43]. Future MWAS studies may compliment the current study and other proteomics studies in the manipulation of therapeutic targets for AD.

A strength of this study is the well-phenotyped clinical characterisation of the discovery cohort, carried out at a national specialist centre, with diagnosis supported by imaging and CSF biomarkers. Although the number of individuals was small, we were able to provide an independent discovery and validation cohort. Unusually we had access to ‘ideal’ healthy controls who were the relatives of patients with AD. This meant that they were age matched, did not have cognitive symptoms, and agreed to have a LP as part of another observational study [44], which allowed us to confirm that they did not have biochemical evidence of brain amyloid pathology. Conversely, the validation cohort had limited clinical information, based on the restrictions of the ethics under which these individuals were recruited and therefore the individuals included could be at a different stage of the disease. However, the results obtained in this work are extremely promising and should be replicated in much larger cohorts of samples from patients across a range of neurodegenerative diseases and across AD disease course, especially during the pre-clinical phase to understand the assay’s potential as a pre-symptomatic screening tool.

In summary, we have identified and validated differences in the urinary proteome between patients diagnosed with AD and healthy age-matched controls. The use of a machine learning approach in combination with a multiplexed panel of biomarkers, allowed us to improve the power of the assay to correctly identify the majority of AD patients analysed in this study. This provides proof of concept that a multiplex of AD-related proteins can be detected in urine using mass spectrometry and demonstrates the potential utility of high-throughput panels of urinary biomarkers to aid clinical diagnosis of AD.

## 4. Materials and Methods

### 4.1. Discovery Cohort

Individuals who fulfilled clinical research criteria for AD were prospectively recruited from a specialist cognitive disorders clinic as previously described [44]. All individuals underwent mini-mental state examination (MMSE; [45]) brain MRI, detailed neuropsychological assessments and had a LP. Controls were prospectively recruited and were age and sex matched (spouses or friends of those with AD), had no cognitive complaints, MMSE scores ≥ 28 and non-AD CSF profiles. Apolipoprotein E (APOE) ε2/ε3/ε4 genotype status was determined as previously described [44].

### 4.2. Validation Cohort

Individuals who fulfilled consensus criteria for AD were prospectively recruited from a diagnostic LP clinic. We included individuals who fulfilled clinical research criteria for AD [46], but CSF and other clinical data were not available for inclusion in this study. Controls were accompanying family or friends and reported no cognitive problems. Only basic demographic data were collected.

### 4.3. CSF Sample Collection, Pre-Analytical Handling, and Analysis

CSF samples were prospectively collected in the discovery cohort according to a standard operating procedure. Aβ1–42, T-tau, and P-tau assays were performed in batches according to local laboratory standard operating procedures to achieve inter-day coefficients of variation (CV) < 10% as previously described [47].

### 4.4. Urine Sample Collection and Pre-Clinical Handling

Individuals were asked to provide a midstream sample of urine during their research visit and to fill a 100 mL polypropylene screw-top container (Starstedt, product code: 75.1354.001). Samples were collected between 0800 and 1200. Samples were collected at room temperature and transferred directly to a −80 freezer within 2 h. 

### 4.5. Urinary Creatinine Measurements

Creatinine concentration was measured for each urine sample using mass spectrometry as previously described [48].

Discovery proteomics: Samples were thawed at room temperature and vortexed for five seconds prior to aliquoting of 2 mL urine into 5 mL centrifuge tubes from Eppendorf. The 2 mL aliquots were centrifuged at room temperature at 3761× *g* for 30 min to separate the urinary sediment from solution using a Sorvall Legend RT centrifuge. The supernatant was transferred to Amicon Ultra-4 10 kDa molecular weight cut-off filters from Merck Millipore (Burlington, MA, USA) and 2 mL Milli-Q water was added to give a final volume of 4 mL. To concentrate the urinary proteins, the samples were centrifuged at room temperature for one hour at 4444× *g* (Sorvall Legend RT). The concentrate was transferred to a 1.5 mL centrifuge tube (Eppendorf). To ensure maximum recovery, the filters were washed with 100 μL 50 mM ammonium bicarbonate which was pooled with the concentrate. 800 μL ice-cold acetone was added to the pooled concentrate and the samples were vortexed for five seconds before overnight incubation in −20 °C. To separate the supernatant from the protein pellet, the samples were centrifuged for 10 min at +4 °C and 16,900× *g* using a Micro-centrifuge 5424 R (Eppendorf). The supernatant was carefully pipetted off and discarded. The pellet was air-dried in a fume hood for 20 min to evaporate residual acetone. 100 μL Milli-Q water was added to the samples and the protein pellet broken up by vigorous vortexing. The samples were thereafter freeze-dried overnight, followed by tryptic digestion with overnight incubation, solid phase extraction to purify the peptides and overnight evaporation of solvents [49]. The digested and SPE-cleaned samples were reconstituted in 50 μL 3% acetonitrile, 0.1% TFA and a peptide assay was performed. The peptide concentration in the samples was normalised to 1000 ng/μL before instrumental proteomics discovery analysis by 2D-LC-MS.

The peptides were separated using a 2D-NanoAquity liquid chromatography system (Waters, Manchester, UK). All samples were fractionated online into ten fractions over a 12-h period as previously described [50]. After acquisition, data were imported to Progenesis QI for proteomics (Waters) and the fractions 1–10 were individually processed before all results were merged into one experiment. The Ion Accounting workflow was utilised, with UniProt Canonical Human Proteome (exported 2017) as database. The digestion enzyme was set as trypsin. Carbamidomethyl on cysteines was set as a fixed modification; deamidation of glutamine and asparagine, oxidation of tryptophan and pyrrolidone carboxylic acid on the N-terminus were set as variable modifications. The identification tolerance was restricted to at least two fragments per peptide, three fragments per protein and one peptide per protein. A false discovery rate of 4% or less was accepted. The individual fractions were combined in Progenesis, using the multifraction experiment workflow. At least two unique peptides per protein and an identification confidence score larger than 15 were set as thresholds for classifying a protein as a confident identification. 

### 4.6. Development of a High-Throughput, Multiplexed and Targeted Proteomic Assay

Potential biomarkers and those proteins demonstrating significant changes in protein expression were further subdivided and ranked into three groups A, B and C. Rank A biomarkers consisted of protein identifications where at least two unique peptides were observed and had a confidence score higher than 15. Other proteins were ranked B by having at least two unique peptides were identified/confidence score higher than 15, and C ranking indicates that one unique peptide was observed and the confidence score was lower than 15. All proteins, regardless of confidence rank, were then included in bioinformatic analyses where we looked for statistically significant changes in expression and fold change. However, only proteins ranked as A and identified by this high stringency analysis, thus more likely to better and more robust biomarkers, were taken forward for validation using a targeted proteomic approach.

The sample cohort for the targeted validation study was prepared using the same method as the discovery cohort but using 4 mL of urine and with 150 ng whole protein ENO1 (yeast) added as an internal standard to account for losses during sample preparation and for quantitation. The concentrated, digested and SPE-cleaned urine samples were reconstituted in 50 μL 3% acetonitrile, 0.1% TFA, containing 0.1 μM of stable heavy isotope labelled peptides or ‘AquaPeptides’ (ALDOA, C3, GSTO1, RSU1 and TSP1) prior to analysis by UPLC-MS/MS. The analytical settings have been described previously [28]. The MRM method consisted of 189 unique peptides and was split over two injections to ensure adequate acquisition of the transitions. A table detailing MRM transitions is available in Appendix A. After acquisition, data were peak picked using an in-house developed Python-based guided user interface (GUI) and peak picking application [51] or the MassLynx (version 4.1) module TargetLynx (Waters). In the GUI application workflow, the raw instrument files were converted to text files using the application MSConvert from ProteoWizard [52] and imported to the application. Peaks were aligned if necessary, and thereafter integrated. When TargetLynx was used, data were imported to the application and quantitative methods were created and applied to the data. The targeted raw data are available via the Panorama repository https://panoramaweb.org/AD_Urine_Proteomics.url (accessed on 27 August 2023). A complete summary of the workflow and methodology is described in Figure 5.

### 4.7. Statistical Analysis

Most of the statistical analyses were performed in Python (version 3.6) [53]. The proteins detected in the discovery analysis transformed to normality using a Box-Cox transformation (boxcox) from SciPy’s stats package (version 1.11.0). Student’s *t*-test with a nominal *p*-value threshold of 0.05 was applied to determine differential protein expression between the groups. Correlation analyses were performed using Pearson correlation (SciPy). The data were visualised using the Seaborn library (version 0.12.2) and the multivariate tool principal component analysis in the software SIMCA, version 17 (Umetrics Sartorius Stedim, Umeå, Sweden).

Linear regression models of proteins analysed in the targeted workflow were built using Scikit-learn (version 1.2.2). Feature selection was performed using a 5-fold cross validated elastic net regression (Scikit-learn) to pinpoint the variables of greatest importance to discriminate between AD and control samples. A multiple linear regression model was built using the panel of selected proteins. Cross validation was performed using KFold cross validations from Scikit-learn with ten splits of the data.

### 4.8. Pathway Analysis

The proteins expressing a nominally significant difference (*p* < 0.05) between the AD and the control groups were investigated using the expression analysis workflow in Ingenuity Pathway Analysis (Qiagen, Venlo, The Netherlands). To determine up- or downregulation of a protein in the AD group, the average fold-change compared to the control group was calculated.

### 4.9. Ethics

Ethical approval was obtained from the National Hospital for Neurology and Neurosurgery Research Ethics Committee (12 LO 1504, Dec 2012, Queen Square ethics committee) and written informed consent was obtained from all participants.

## Figures and Tables

**Figure 1 ijms-24-13758-f001:**
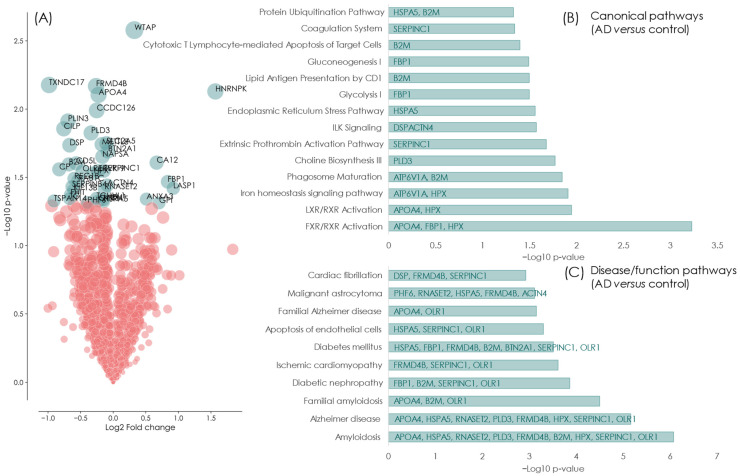
Discovery urine proteomics of AD versus healthy age-matched controls. (**A**) Volcano plot showing the nominally differentially expressed (*p* < 0.05) proteins in blue and non-significantly different proteins in red. Ingenuity pathway analyses of those proteins observed to have differential expression in the urine of AD patients versus normal, age-matched controls; (**B**) Canonical pathway analysis demonstrating that significant differences are observed in lipid and cholesterol homeostasis which is a hallmark of AD; (**C**) Disease and functional analyses clearly identifies that there is significant changes in biomarkers involved in amyloidosis and AD, indicating that proteins present in the urine identify underlying disease processes occurring in the brain.

**Figure 2 ijms-24-13758-f002:**
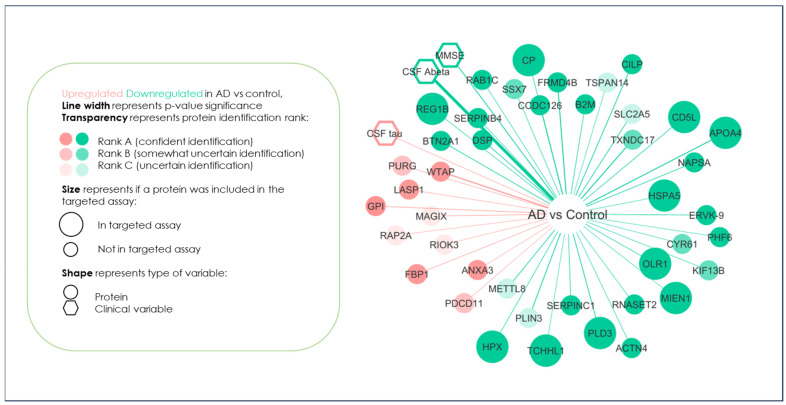
Summary of the biomarker discovery experiment and the identification of those proteins to be put forward as evaluation as potential biomarkers capable of distinguishing AD from controls. Proteins identified with high confidence and amenable to tandem mass spectral analyses are shown as **large green circles** (downregulated in AD) and **pink circles** (upregulated in AD).

**Figure 3 ijms-24-13758-f003:**
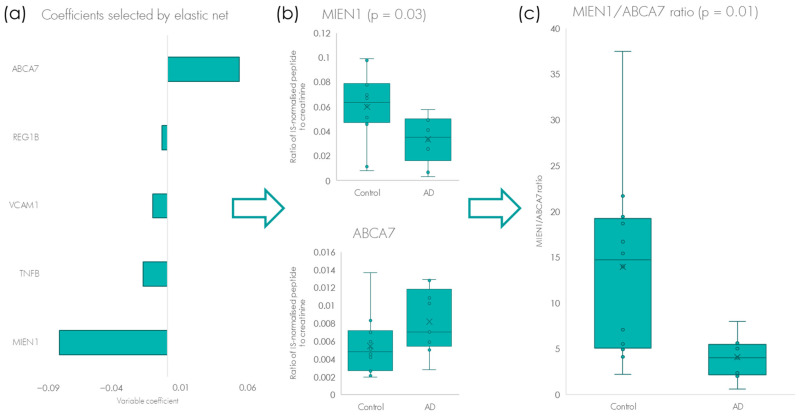
(**a**) Identification of the five proteins that demonstrated significant elevation or were present in lower amounts in the urine of patients with a confirmed AD diagnosis (MIEN1, TNF-beta, VCAM1, REG1B and ABCA7). (**b**) Univariate analyses of the two proteins most up- and down regulated, ABCA7 and MIEN1, respectively, with MIEN1 being statistically significantly elevated (*p* = 0.03). (**c**) By looking at the ratio of the biomarkers ABAC7 and MIEN1 to one another, increases the statistical significance between AD and controls even further (*p* = 0.01). The whiskers show the minimum and maximum and the boxes show the 25th percentile, the median and the 75th percentile. Values outside 1.5 fold the interquartile are represented by dots. x represents the mean.

**Figure 4 ijms-24-13758-f004:**
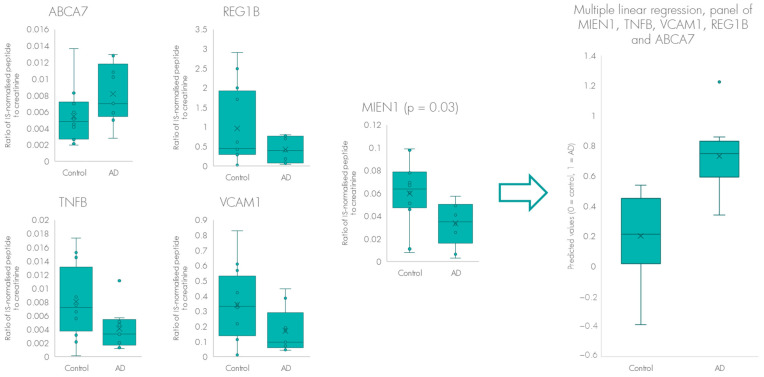
Figure showing the power of multilinear regression and multiplex biomarker panel analyses to increase accuracy and sensitivity of the test by using four biomarkers in addition to the best performing biomarker (MIEN1).

**Figure 5 ijms-24-13758-f005:**
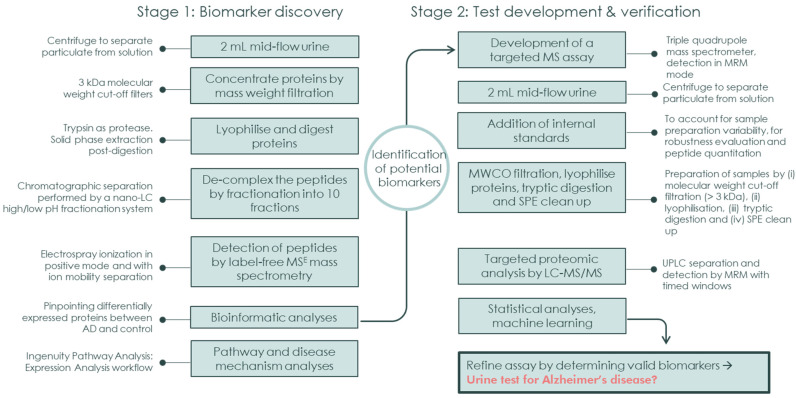
Methodology Workflow: shows methodology workflow for the discovery cohort protein analysis (stage 1) and the development of a high-throughput, multiplexed and targeted proteomic assay (stage 2).

**Table 1 ijms-24-13758-t001:** (**a**): Demographics, cognitive profiles and cerebrospinal fluid biomarker profiles of the discovery cohort. Abbreviations: AD, Alzheimer’s disease; MMSE, mini mental score exam; CSF, cerebrospinal fluid. All data are shown as the mean ± standard deviation unless otherwise stated. (**b**): Demographics and cognitive profiles of the validation cohort. Abbreviations: AD, Alzheimer’s disease. All data are shown as the mean ± standard deviation unless otherwise stated.

(**a**)		
	**AD (n = 6)**	**Control (n = 5)**
**Sex (% Male)**	50	75
**Age (years)**	59.2 ± 4.1	59.3 ± 3.9
**Positive APO** ε**4 status (%)**	83	Not tested
**MMSE score**	24.2 ± 3.5	29.4 ± 0.9
**CSF Aβ1-42 (pg/mL)**	453 ± 93.4	1073.2 ± 196.5
**CSF T-tau (pg/mL)**	1407 ± 985.3	304 ± 80.1
**CSF P-tau 181 (pg/mL)**	121.4 ±89.5	42.4 ± 8.0
**BMI**	24.1 ± 2.3	25.98 ± 2.2
(**b**)		
	**AD (N = 9)**	**Control (N = 12)**
**Sex (% Male)**	67	50
**Age (years)**	62.3 ± 3.0	59.1 ± 6.6

## Data Availability

The targeted raw data are available via the Panorama repository https://panoramaweb.org/AD_Urine_Proteomics.url (accessed on 27 August 2023). Any further data is available from the corresponding author by reasonable request.

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
