# Peer review of "A Multiplexed Urinary Biomarker Panel Has Potential for Alzheimer’s Disease Diagnosis Using Targeted Proteomics and Machine Learning"

_ijms, 2023, doi:10.3390/ijms241813758_

Round 1

Reviewer 1 Report

Dear authors,

the manuscript describes a potential novel biomarker for the diagnosis of Alzheimer's disease using a non-invasive clinical methodology as opposed to currently used methods. The results are promising and the manuscript has scientific quality, however some suggestions are made.

The introduction is complete and detailed and does not require major changes. Some bibliographical reference could be added on the physiology of Alzheimer's disease 10.3390/biomedicines9121910.

The results are displayed adequately and the figures are easy to interpret. Perhaps the statistical test used by which significant differences have been obtained could be indicated (figures 4 and 5). In addition, these figures do not include the units of measurement represented on the Y axis (figure 4b, 4c and figure 5).

In the discussion it would be interesting to include the biomarkers found in the study and the relationship of their functionality in Alzheimer's disease. Why do the authors think that most biomarkers are related to metabolism? Are the authors aware of any relationship between changes in lipid metabolism and degree of disease?

No grammatical error has been detected

Author Response

Reviewer 1

the manuscript describes a potential novel biomarker for the diagnosis of Alzheimer's disease using a non-invasive clinical methodology as opposed to currently used methods. The results are promising and the manuscript has scientific quality, however some suggestions are made.

We thank the reviewer for their positive comments and for recognizing the novelty and value of our mass spectrometry test.

The introduction is complete and detailed and does not require major changes. Some bibliographical reference could be added on the physiology of Alzheimer's disease 10.3390/biomedicines9121910.

We thank the reviewer for this suggestion and had now added the references (Garcia-Morales, 2019) as suggested.

The results are displayed adequately and the figures are easy to interpret. Perhaps the statistical test used by which significant differences have been obtained could be indicated (figures 4 and 5). In addition, these figures do not include the units of measurement represented on the Y axis (figure 4b, 4c and figure 5).

We thank the reviewer for this suggestion and have added the significance test to the figure legend. We also want to thank the reviewer for bringing the lack of units of the box plots to our attention. We have updated the figures so that they now contain a description on the axes.

In the discussion it would be interesting to include the biomarkers found in the study and the relationship of their functionality in Alzheimer's disease. Why do the authors think that most biomarkers are related to metabolism?

That is a very good question - we are not sure, but there are a significant number of references in relation to lipid metabolism, including cholesterol, phospholipids, glycosphingolipids and disturbances in lysosomal function in AD.

Are the authors aware of any relationship between changes in lipid metabolism and degree of disease?

The authors are unaware of any lipid biomarkers in urine that could be used to stratify disease progression or diagnose AD. This is however a very good point and a lipidomic profiling of urine could identify new potential biomarkers. Unfortunately, this is outside of the scope of this manuscript.

Reviewer 2 Report

The authors investigate the differential proteomic of urine from AD patients and controls and find one protein MIEN1 is significantly decreased in AD patients, compared with healthy control. 

The impact of the current study is limited. as previous similar studies have been reported. In a recent study (Dement Geriatr Cogn Dis Extra. 2019 Jan-Apr; 9(1): 53–65), 109 differential presented proteins are identified in AD patient urines, compared with healthy controls. Furthermore the MIEN1 is not included in the differential protein list. The discrepancy of results from the current study and previous reports may be due to the less amount of subjected recruited (6 AD patients) in the current study. The authors have also not identified the disease stage of recruited patients (early stage or late stage AD?). This can be another reason for limited positive findings from their study. Therefore the clinical significance of the current study will be limited. 

Furthermore to be more convincing, the authors should perform western blot analysis of MIEN1 in AD and control urine, as this protein is the only positive findings of their study. 

Author Response

Reviewer 2

The impact of the current study is limited. as previous similar studies have been reported. In a recent study (Dement Geriatr Cogn Dis Extra. 2019 Jan-Apr; 9(1): 53–65), 109 differential presented proteins are identified in AD patient urines, compared with healthy controls. Furthermore the MIEN1 is not included in the differential protein list. The discrepancy of results from the current study and previous reports may be due to the less amount of subjected recruited (6 AD patients) in the current study.

This was an error on our part and has now been addressed. We thank the reviewer for pointing out that MIEN1 was indeed differentially expressed in both the untargeted and targeted studies. We have further clarified this by including a supplementary table with the significantly differentially expressed proteins identified in the discovery study. We apologise for this mistake. In regards to the comparison between the two studies, possible explanations for the differences in number of significant proteins could be differences in instrumentation (e.g. some peptides ionise different depending on the technology), differences in identification stringency, and – as the reviewer suggests – differences in sample numbers. Another explanation could be that our controls samples are genuine controls. We have been able to biologically classify these individuals as not having AD pathology based on CSF biomarkers, rather than classifying them neuropsychologically, based on MMSE alone.

The authors have also not identified the disease stage of recruited patients (early stage or late stage AD?). This can be another reason for limited positive findings from their study. Therefore the clinical significance of the current study will be limited.

We have provided the MMSE scores of the patients in the demographics table. To provide further clarity we have stated in the results section “with individuals in the mild to moderate stage of the disease”. We did not have access to detailed clinical information of the validation cohort. We have previously declared this as a limitation of the study, but now further emphasize this by adding the following sentence: “and therefore the individuals included could be at a different stage of the disease”.

Furthermore to be more convincing, the authors should perform western blot analysis of MIEN1 in AD and control urine, as this protein is the only positive findings of their study.

We accept that this would be a valuable addition to the study but we have no remaining samples on which to carry out a Western blot analysis. We are planning to extend this project and carry out a further analysis on a much larger cohort of well characterized patients, but this is beyond the scope of this manuscript. However, targeted proteomics is the gold standard for protein identification and quantitation as it identifies proteins according to peptide sequence, parent, daughter ion fragmentation and LC retention time this providing 3 parameters of specificity and making it the premier protein quantitation tool. In contrast Western Blotting is significantly less sophisticated compared to targeted proteomics regarding identification and especially quantitation. To confirm MIEN1’s identity unequivocally, we had synthesised commercially two peptide standards used in this study as markers for the protein. This then included the matching of corresponding retention times between MIEN1 standards observed technically and what was endogenously present in the patient samples which matched identically and validated the identification of MIEN1. In summary, we have used the gold standard method for quantitation of proteins and is recognised as being superior to Western blotting.

Round 2

Reviewer 2 Report

The new added data is interesting which provide more information for readers. 

However, both 2 tables are entitled with Table S2 which need to be corrected.  Furthermore, for the supplementary table 2 of Significantly different proteins in the comparison of urine protein expression between AD and healthy controls, the table should have separating lines as another supplementary table. 

Author Response

However, both 2 tables are entitled with Table S2 which need to be corrected. 

Apologies - now corrected 

Furthermore, for the supplementary table 2 of Significantly different proteins in the comparison of urine protein expression between AD and healthy controls, the table should have separating lines as another supplementary table

The table has been reformatted to add top horizontal lines, to match table S2. 
